# Gas Chromatography-Mass Spectrometry and Single Nucleotide Polymorphism-Genotype-By-Sequencing Analyses Reveal the Bean Chemical Profiles and Relatedness of *Coffea canephora* Genotypes in Nigeria

**DOI:** 10.3390/plants8100425

**Published:** 2019-10-18

**Authors:** Chinyere F. Anagbogu, Christopher O. Ilori, Ranjana Bhattacharjee, Olufemi O. Olaniyi, Diane M. Beckles

**Affiliations:** 1Department of Plant Sciences, University of California, Davis, CA 95616, USA; dmbeckles@ucdavis.edu; 2Department of Crop Protection and Environmental Biology, University of Ibadan, Ibadan 200284, Nigeria; chrisilori@yahoo.com; 3Crop Improvement Division, Cocoa Research Institute of Nigeria, Ibadan 200255, Nigeria; yinkamoji@yahoo.com; 4Bioscience Laboratory, International Institute of Tropical Agriculture, Ibadan 200285, Nigeria; r.bhattacharjee@cgiar.org

**Keywords:** *Coffea canephora*, metabolomics, genomics, cup quality, health benefits

## Abstract

The flavor and health benefits of coffee (*Coffea* spp.) are derived from the metabolites that accumulate in the mature bean. However, the chemical profiles of many *C. canephora* genotypes remain unknown, even as the production of these coffee types increases globally. Therefore, we used Gas Chromatography-Mass Spectrophotometry to determine the chemical composition of *C. canephora* genotypes in Nigeria—those conserved in germplasm repositories and those cultivated by farmers. GC-MS revealed 340 metabolites in the ripe beans, with 66 metabolites differing (*p*-value < 0.05) across the represented group. Univariate and multivariate approaches showed that the ‘Niaouli’ genotypes could be clearly distinguished from ‘Kouillou’ and ‘Java’ genotypes, while there was almost no distinction between ‘Kouillou’ and ‘Java,’. Varietal genotyping based on bean metabolite profiling was synchronous with that based on genome-wide Single Nucleotide Polymorphism analysis. Across genotypes, the sucrose-to-caffeine ratio was low, a characteristic indicative of low cup quality. The sucrose-to-caffeine ratio was also highly correlated, indicative of common mechanisms regulating the accumulation of these compounds. Nevertheless, this strong correlative link was broken within the ‘Niaouli’ group, as caffeine and sucrose content were highly variable among these genotypes. These ‘Niaouli’ genotypes could therefore serve as useful germplasm for starting a Nigerian *C. canephora* quality improvement breeding program.

## 1. Introduction

Coffee (*Coffea* spp.) is one of the most consumed beverages globally, and there is a clear correlation between the sensory quality of a coffee variety and its market value [1]. Although *Coffea canephora* does not have the cup quality of the more popular *Coffea arabica*, it continues to be widely grown, especially in regions where farming is low intensive because of its relative tolerance to a range of biotic and abiotic stresses [2]. As a result of this, and its competitive pricing, the *C. canephora* market share has increased in recent years, totaling 40% of the coffee traded globally [3]. Importantly, *C. canephora* forms a critical source of income for millions of smallholder coffee farmers in developing countries [3]. In Nigeria, it accounts for 90% of coffee production [4]. Since quality is a major selection criterion for coffee improvement [5], it is important to determine the chemical profiles of *C. canephora* varieties which are important to many resource-poor farmers.

Little is known about the genetic background and the quality profile of the genotypes used for coffee cultivation in Nigeria. The early establishment of a *C. canephora* polyclonal seed garden in Nigeria was based on five sexually compatible genotypes of ‘Java Robusta’, ‘Kouillou’ and ‘Niaouli’ varieties at the Cocoa Research Institute of Nigeria (CRIN), for replication and distribution to farmers [6]. The use of these varieties led to an increase in coffee yields from 800 kg to 1.4 ton per hectare [6]. We recently used Single Nucleotide Polymorphism-Genotype-By-Sequencing analyses to determine the relatedness of some of the conserved and cultivated *C. canephora* genotypes in Nigeria [7]. The data showed that the genetic diversity of the conserved CRIN germplasm was narrow and comprised of three genetic groups (‘Niaouli’, ‘Kouillou’ and ‘Java Robusta’)—of which, one of the groups (‘Niaouli’) dominated the cultivated accessions [7]. The chemical composition of the beans from these genotypes has yet to be investigated. Therefore, their sensory profiles are unknown.

Understanding the metabolomics profile of coffee is essential for coffee cup quality improvement. Metabolite profiling has been used to discern coffees of different origins, e.g., Asia, Africa, and South America [8], and in linking high grade Colombian *C. arabica* beans to its higher sucrose content [9]. Metabolite profiling is also the most efficient way to determine the relative level of key chemicals in coffee that affect quality [10]. Several compounds with important contributions to the complex chemistry of coffee roasting have already been identified [11,12]. For example, two abundant metabolites, caffeine and chlorogenic acid at high amounts, contribute to bitterness and lead to low cup quality [13,14]. Among the fatty acids, linoleic and palmitic acids correlate with high-quality coffee, while oleic and stearic acids are associated with inferior flavors [15]. In *C. arabica,* many of the alcohols, aldehydes, hydrocarbons, and ketones present in the beans are associated with poor aroma, volatility and acidic profile, causing an earthy flavor [16].

While the metabolite profiles of coffee beans from various regions are well represented in literature [8], the chemical composition of the beans of *C. canephora* genotypes in Nigeria is unknown, and their quality from a chemical perspective is lacking. A combination of genome-wide Single Nucleotide Polymorphisms and metabolomic analyses could help to link variation in genotype to variation in phenotype (mainly quality parameters). Some Single Nucleotide Polymorphisms (SNPs) may even cause significant changes in the accumulation of some metabolites, which may be manifested at the phenotypic level as alterations in disease resistance and cup quality [5,17]. Different analytical techniques such as High-Performance Liquid Chromatography (HPLC) [18,19], Gas Chromatograph-Mass Spectrometry (GC-MS) and Proton Nuclear Magnetic Resonance (H^1^-NMR) [20] have been successfully used for the classification of coffee genotypes. Metabolomics of coffee beans will help to decipher the features contributing to good quality and their profiles, while the genomics study will help in linking good quality metabolomic profiles to DNA polymorphisms.

The aim of this study is to determine the chemical fingerprint of the coffee germplasm maintained at CRIN and those grown by Nigerian farmers, and to relate this to their SNP-variation. Specifically, we aim to: (a) assess the metabolomic profiles of *C. canephora* genotypes cultivated in Nigeria, (b) determine the chemical diversity of the genotypes in relation to SNP diversity and (c) identify genotypes with potential good cup quality to be incorporated in the breeding program for cup quality improvement.

## 2. Results and Discussion

### 2.1. Metabolite Profiles of the Three C. canephora Genotypes: “Niaouli’, ‘Kouilou’ and ‘Java Robusta’

A total of 340 untargeted metabolites were detected—of which, 163 (Appendix A) were identified across all three varieties of *C. canephora*, ‘Niaouli’, ‘Kouilou’ and ‘Java Robusta’, used in the study. A similar number of metabolites (182) was detected in Asian palm civet coffee using GC-MS [21]. The metabolites identified in our study were grouped into eight biochemical classes: amines, amino acids, sugars, sugar derivatives, organic acids, fatty acids, phenolic acids/alkaloids and inorganic compound (Appendix A).

The major compounds identified in the Nigerian varieties were compared to those in other studies. Caffeine, chlorogenic acids, quinic acid, citric acid and sucrose were the most abundant metabolites identified in these varieties (Figure 1), which is similar to the predominant compounds identified in *C. arabica* derived from Asian palm civet [21]. Stearic acid, palmitic acid, linoleic acid and pelargonic acid were the major fatty acids in the three Nigerian genotypes (Appendix A). This fatty acid profile was similar to that described by Dong et al., who studied the green coffee beans of seven cultivars of *C. canephora* grown in Hainan Province, China [22]. In their study, and similar to ours, linoleic acid and palmitic acid were among the most abundant fatty acids, but oleic acid, and arachidic acid were also high in content [22]. The most abundant amino acids in our study did not overlap with those identified by Dong et al. [22]. In this study, aspartic acid, glutamic acid, proline, and tryptophan were highest in abundance (Appendix A), while Dong et al. identified leucine, lysine and arginine to be the most dominant amino acids in their study [22].

Next, we looked at the compounds known to influence coffee quality. The relative abundance of important cup quality precursors, including amino acids, fatty acids, and sugars was low, while the compounds associated with poor hedonistic value, such as caffeine, organic acids, and polyphenols were relatively high (Figure 1). This suggested that the sensory attributes of these Nigerian varieties may not match that of high-quality coffees [14,19,23,24] and that there is a need for a concerted coffee breeding program in Nigeria focusing on the improvement of sensory attributes.

Another important aim was to look for variability in metabolites among the Nigerian genotypes. One-way ANOVA identified 66 (~20%) metabolites that differed among the coffee varieties within the ‘Niaouli’ group (Nia_1, Nia_2 and Nia_3), the ‘Kouillou’ group (C111 and C36) and the ‘Java Robusta’ (T1049) group (*p* < 0.01 and FDR < 0.05; Table 1 and Appendix A). Many metabolites could be differentiated between the ‘Niaouli,’ and, the ‘Kouillou’ genotypes, and between the ‘Niaouli,’ and ‘Java Robusta’ (herein called ‘Java’) genotypes. However, in contrast, there were few differences in metabolite abundance between ‘Kouillou’ and ‘Java’. As a result, the latter two genotypes were grouped together and described as ‘Kouillou/Java.’

### 2.2. Metabolomic Markers for Differentiating Genotypes

Metabolomic markers can be useful in differentiating genotypes grown in different regions and in selecting genotypes with metabolite profiles for use in breeding programs [10,25]. Such metabolites were identified in this study by using the analysis of variance (ANOVA), and more potential markers were uncovered using variable importance in projection (VIP) at a Partial Least Square (PLS) regression value above 2 (Figure 2). Most of the discriminatory metabolomic markers were in abundance in ‘Kouillou/Java’ compared to ‘Niaouli’ (Table 1; Figure 2). These distinct metabolites were mainly sugar derivatives, while the others were organic acids, amino acids and amines. The most distinctive metabolomic marker differentiating ‘Niaouli’ from ‘‘Kouillou/Java’ was an unknown compound ‘6404,’ identified using both the Significant Analysis of Microarrays (SAM) and one-way ANOVA analyses. It was higher in ‘Kouilou/Java’ than ‘Niaouli’. However, its role in flavor characteristics is unknown and needs to be further studied.

Sugar derivatives. These are compounds that are usually derived from monosaccharides, but which have undergone further chemical modification. They include sugar alcohols, sugar acids, amino sugars and deoxy sugars. Of these, the sugar alcohol palatinitol was higher in ‘Kouillou/Java’ compared to ‘Niaouli’ (Table 1; Figure 2) and could therefore serve as a potential biomarker to differentiate between these genotypes. The farmers’ cultivated accessions, which were all of the ‘Niaouli’ group, had low palatinitol content, and this metabolite can be used to distinguish among *C. canephora* varieties in Nigeria (*p* = 1.30 × 10^−6^).

Amino acids. Generally, amino acids make positive contributions to coffee hedonistic value. They react favorably with sugars during the Maillard reaction, producing pleasant aroma precursors [26]. Tryptophan, proline, and threonine were the amino acids also identified as potential biomarkers. Threonine and proline have high-quality attributes [27], while tryptophan is linked to low bean quality; it is a specific marker linked with bean immaturity, and the presence of high levels negatively impact the flavor of roasted coffee [28] Tryptophan levels were higher, while proline and threonine were lower (*p* = 3.88 × 10^−8^), in ‘Kouillou/Java’ compared to ‘Niaouli’. This amino acid profile suggested that ‘Kouillou/Java’ has an inferior amino acid profile compared to ‘Niaouli’ (Table 1). Notably, the farmers’ accessions, which were all of the ‘Niaouli’ genotypes, were characterized by high threonine and proline with *p*-values of 1.68 × 10^−5^ and 5.27 × 10^−7^, respectively.

Polyphenols. Coffee is one of the most important sources of polyphenols. The polyphenols with the potential to act as biomarkers of Nigerian *C. canephora* coffee beans were tyrosol, gluconic acid and nornicotine. Tyrosol and nornicotine were high in ‘Niaouli’ compared to ‘Kouillou/Java’ (Table 1). Tyrosol is an essential polyphenol in olive oil which protects the actin filament network from oxidized Low-Density Lipoproteins. The tyrosol derived from coffee could also serve a protective function [29].

Fatty acids. In coffee beans, fatty acids contribute to the desirable aroma precursors of roasted coffee beans [30] but high fatty acids metabolites are linked to low grade (quality) coffee [29]. We found that hexadecyl glycerol and arachidic acid were statistically higher in ‘‘Kouillou/Java’ compared to ‘Niaouli’. Interestingly, glycerol was one of the few metabolites that could differentiate between ‘Java’ and ‘Kouillou’ and it was higher in the latter, compared to the former (Table 1).

### 2.3. Metabolite-to-Metabolite Correlations and Their Potential Influence on Cup Quality and Other Beneficial Traits

The relative abundance of multiple metabolites and their interaction greatly affects cup quality and other beneficial properties of coffee [31]. To study these interactions, heatmap analyses of metabolite-to-metabolite correlations were drawn, identifying metabolites that are positively and negatively correlated in abundance across genotypes (Figure 3a,b).

Caffeine. This metabolite correlated moderately with sucrose (*r*^2^ = 0.42; Table 2). This moderate association of sucrose with caffeine should have the effect of reducing the bitterness caused by the high caffeine content of many *C. canephora* types [31]. Although caffeine contributes to bitterness in coffee, it has long been valued for its beneficial cognitive and other health effects [32], even whether there are negative consequences for overconsumption [33].

Sugars. Sucrose and fructose were the two major sugars found in the coffee beans of *C. canephora* cultivated in Nigeria (Appendix A). High-quality coffee usually accumulates high levels of sucrose [9]. The positive Pearson’s correlation coefficient for sucrose and caffeine observed in this study was in contradiction to the negative correlation found by Caporaso et al. [34]. In our work, sucrose and caffeine were found to be positively correlated with two unknown compounds ‘16548’ and ‘68’ (Table 2). Interestingly, a negative correlation was found between sucrose and palatinitol (Table 2); this could be explained by the fact that sucrose is a precursor of palatinitol biosynthesis [35]. Metabolite ‘6404’ is potentially a metabolomic marker, differentiating cultivated accession ‘Niaouli’ from ‘Java/Kouillou’ (Table 1). This metabolite could contribute to low cup quality as it was negatively correlated with sucrose (*r* = −0.37).

Phenolic acids. Chlorogenic acid and quinic acid were the most abundant polyphenols (Appendix A). These compounds cause a high degree of bitterness and could contribute to the low cup quality [36] of Nigerian *C. canephora*, since they are higher in abundance relative to sucrose, amino acids and the fatty acids, compounds that are associated with high cup quality [27,37,38]. However, while chlorogenic acids negatively affect taste, they do have beneficial health properties as antioxidants.

Amino acids. Several amino acids identified as good cup quality precursors were found to be positively associated in coffee genotypes used in the study. These included aspartic acid, glutamine, oxoproline, serine, N-acetyl-D-galactosamine, beta-glutamic acid, proline and threonine (Figure 3a). Of these, it is known that proline produces pleasant, flowery and fragrant aromas; aspartic acid and serine generates pleasant and fruity aromas; and threonine produces a pleasant caramel-like odor [27]. Aspartic acid and glutamic acid could be explored as potential markers differentiating Nigerian *C. canephora* genotypes. These amino acids accumulated to relatively high levels (Appendix A) compared to that reported by Arnold et al., [39], where asparagine and glutamic acid were the two major amino acids found in the coffee beans in their study.

Fatty Acids. Arachidic acid, stearic acid, palmitic acid, linoleic acid and glycerol are fatty acids linked to coffee with better sensory qualities [30] and, in this study, we found these metabolites to be positively correlated with each other (Appendix A).

Sugar alcohols. Apart from cup quality improvement, the identification of genotypes with high drought tolerance traits is of great importance in the face of climate change. Here, galactinol and beta-gentiobiose were positively and highly correlated, (*r*^2^ = 0.98) (Figure 3b). Galactinol, together with raffinose, are essential in protecting plant cells from the oxidative damage caused by various types of stress conditions [40]. Galactinol and beta-gentiobiose had the highest positive correlation coefficient (*r*^2^ = 0.97), followed by lactobionic acid and beta-gentiobiose (*r*^2^ = 0.96). Galactinol synthase is a key enzyme in the synthesis of the raffinose family of oligosaccharides, which function as osmoprotectants in plant cells [40]. The Pearson correlation was statistically significant (*p* < 0.05) among different chemicals.

### 2.4. Metabolite and Genomic Diversity within and among Varieties

Principal Component Analysis (PCA) and Partial Least Squares (PLS) both detected group differences. PCA is an unsupervised approach. PLS, in contrast, is a supervised approach that seeks to maximize separation between groups by reducing within group variation [41,42]. This explains why clear separation between ‘Java’ and ‘Kouillou’ was possible with Partial Least Squares-Discriminant Analysis (PLS-DA), but not with PCA (Figure 4). Both PCA and PLS detected 32.2% and 28.7% variations, respectively (Figure 4a,b). They also indicated that there are two metabolomic diverse groups, which we designated as clusters I and II. Cluster I comprised of all the ‘Niaouli’ genotypes, while cluster II was made up of the ‘Java’ and ‘Kouillou’ genotypes (Figure 4a,b).

Because of post-transcriptional modification, a change in metabolite level may result in phenotypic changes more than alterations in DNA sequence [10,43]. Still, it was of interest to compare how the genotypes would be classified based on SNPs vs. based on their metabolite profiles. Genomic analysis through the characterization of 433,048 SNPs grouped the genotypes into three genetic units (clusters III, IV and VI) (Figure 5a). Hierarchical analysis of the GC-MS data also led to a similar clustering of the genotypes i.e., (I, II and III) (Figure 5a). Among the coffee genotypes studied, genetic diversity based on SNPs can be linked to chemical or metabolomics diversity, as the data proved to be complementary. Single Nucleotide Polymorphism analysis of both the phenolic and vitamin E pathways revealed metabolite-specific genetic diversity among the rice varieties examined [44].

### 2.5. Genotypes with Favorable Bean Quality Traits

Our analysis showed that some genotypes contained high and low levels of some important chemicals linked to coffee quality and health benefits (Table 3). High levels of sucrose and low levels of aminobutyric acid (GABA), quinic acid, choline, acetic acid and fatty acids were reported to be the metabolomic markers of high-grade green coffee [9]. The Nia_14 and Nia_15 genotypes of ‘Niaouli’ have a low caffeine content and relatively higher sucrose content (Table 3). Thus, bitterness may be reduced within these genotypes, and the caffeine level may make it suitable to some consumers who have health concerns. Determining whether there are SNP differences in the gene coding for these “favorable” metabolites may point to a genomic cause for differences in metabolite levels among these genotypes (Table 3) that might be exploited in coffee breeding and coffee functional genomic analysis. Sucrose and caffeine showed a broad range of concentrations (Table 3 and Figure 6). The variability of sucrose was 39-fold, while caffeine was 5-fold, offering targets for breeding programs. A high level of natural variability for caffeine and sucrose in green coffee beans was also observed in other studies [24,33]. Metabolites differentiating high- (*C. arabica*) and low-quality coffee (*C. canephora*) are summarized in Table 4.

Levels of sucrose are exceptionally higher in the ‘Niaouli’ genotypes, with the highest concentration found in Nia_25 and Nia_22, while those with the lowest concentration were C111_1 and C36_5. It could be deduced that the genetic group ‘Nia_2’ has high sucrose synthase enzyme and could be linked to SNPs changes. A low sucrose concentration in *C. canephora* green coffee beans has been attributed to the limited capacity of sucrose synthase to synthesize sucrose at the final stage of coffee grain development [45,46,47]. It will be interesting to further evaluate the sucrose synthase sequence of these genotypes having high and low sucrose content to determine whether there is a SNP change at this locus.

## 3. Materials and Methods

### 3.1. Single Nucleotide Polymorphism Genotype-By-Sequencing Analysis

DNA from 48 samples comprising 18 accessions from Cocoa Research Institute of Nigeria (CRIN) and 30 accessions from farmers’ fields were analyzed. The CRIN accessions consisted of five *Coffea* species including *Coffea arabica*, *Coffea abeokutae*, *Coffea liberica*, *Coffea stenophylla* and *Coffea canephora*. The *C. canephora* varieties used were ‘Kouillou’, ‘Gold Coast’, ‘Java Robusta’, ‘Niaouli’, ‘Uganda’ and ‘Java Robusta Ex Gamba’. More information about the methods used for this analysis is available [7].

The farmers’ accessions known to be the ‘Niaouli’ variety were grouped into 3 genotypes (Nia_1, Nia_2 and Nia_3) based on the hierarchical clustering of 433,650 SNPs using SNPRelate software (http://www.Rproject.org) [48]. These genotypes and those within the ‘Kouillou’ and ‘Java Robusta’ groups were used for metabolomics analysis. These varieties (‘Niaouli’, ‘Kouillou’ and ‘Java Robusta’) represented three genetic structures detected by GBS-SNP analysis on *C. canephora* repository in Nigeria [7]. These genotypes were selected because they were used to establish the pioneer model polyclonal seed garden plot, which will be replicated in farmers’ fields.

### 3.2. GC-MS Analysis

The genotypes used for metabolomics were cultivated *C. canephora* accessions in Nigeria [7], consisting of ‘Niaouli’, ‘Kouillou’ and ‘Java Robusta’ varieties. They were further classified into one of six groups based on the result from SNP-GBS analysis [7]. ‘Niaouli’ is comprised of three genotypes: Nia_1, Nia_2 and Nia_3, (classified as Groups 1, 2 and 3, respectively), ‘Kouillou’ is comprised of two genotypes: C111 and C36, (classified as Groups 4, and 5, respectively) and, ‘Java Robusta’ is comprised of one genotype: T1049, and was classified as Group 6. There were five replicates for each group (genotypes), giving a total of 30 samples (Table 5).

Reddish matured (ripened), coffee bean (Figure 7b) of these genotypes were collected in ice bags and immediately transferred to −80 °C. The endosperms of the coffee bean (Figure 7c) were excised using a sterile blade and re-transferred to −80 °C. These endosperms were lyophilized, ground into powder with Udy mill (Udy Corporation) and sealed prior to metabolomics analysis.

The metabolomics analysis was performed according to Fiehn et al. [49]. The analyte was extracted from the sample using a solvent containing isopropanol/acetonitrile/water at the volume ratio of 3:3:2. The supernatant was concentrated in a Centrivap cold trap vacuum concentrator (http://www.labconco.com) at room temperature for 4 h. The extracts were immediately derivatized for GC–Time-of-Flight (TOF) mass spectrometry analysis by adding 90 µL of N-methyl-N-trimethylsilyltrifluoroacetamide and 1% (v/v) trimethylchlorosilane (1 mL bottles; Pierce) to the extract and shaken at 37 °C for 30 min. The reaction mixture was transferred to a 2 mL clear glass auto-sampler vial with micro-insert (Agilent; http://www.agilent.com) and closed using an 11 mm T/S/T crimp cap.

Samples (0.5 μL) were injected between 2 and 24 h after derivatization in an Agilent 6890 gas chromatograph controlled by using LecoChromaTOF software version 2.32 (http://www.leco.com). The analytical GC column was Restek corporation Rtx-5Sil MS (30 m length × 0.25 mm internal diameter with 0.25 μm film made of dimethyl/diphenylpolysiloxane at the volume ratio 95:5) and the mobile phase used was Helium.

The data was acquired on a Mass spectrometry Instrument, a Leco Pegasus IV time-of-flight mass spectrometer controlled using LecoChromaTOF software version 2.32, at a mass resolving power R = 600 from *m*/*z* 85–500 at 20 spectra per second, and 1550 V detector voltage, without turning on the mass defect option. Recording ended after 1200 s.

### 3.3. Statistical Analysis of Metabolomics Data

All metabolites, including those identified and those not identified using the National Institute of Standards and Technology mass spectral library, were used for data analysis. Both univariate and multivariate statistical approaches were performed with metaboAnalystR [50]. One-way Analysis of Variance (ANOVA) test was performed to ascertain the significant variables, and they were expressed as *f*- and *p*-values. The level of statistical significance (−Log10(*p*)) was determined, followed by post-hoc analyses to correct the *p*-value and thus generate the False Discovery Rate (FDR). Fisher’s least significant difference method (LSD) was used to identify groups that differ in their metabolite profile. Pearson’s correlation coefficient between the metabolites was calculated using metaboAnalystR [50]. Principal Component Analysis (PCA) and Partial Least Squares-Discriminant Analysis (PLS-DA) were performed using the prcomp package within the R statistical package [51]. Prominent potential discriminatory metabolites were identified with variable importance in projection (VIP) scores at a PLS regression value above 2. Calculations for these analyses were based on singular value decomposition [50]. Hierarchical Clustering Analysis was used to classify genotypes by employing Pearson’s correlation as a similarity measure. The clustering algorithm used was Ward’s linkage, which minimizes the sum of squares of any two clusters [50]. The metabolomic analyses results were compared to hierarchical clustering of SNP data obtained by SNP relate [50].

## 4. Conclusions

For the first time, metabolites in the beans of Nigerian *C. canephora* were evaluated. The farmers’ cultivated accessions have unique metabolite profiles and genomic structures compared to the conserved accessions. There was high variability within the varieties for caffeine and sucrose—the two key compounds in coffee. However, the high caffeine, quinic acid and chlorogenic acid may contribute to low quality and bitterness. Genomic technologies such as transgenesis, molecular marker-assisted breeding, genomics, proteomics, and metabolomics should complement traditional breeding efforts for hastening the genetic improvement of coffee. The alteration of major metabolites that contribute to coffee taste, such as caffeine and sucrose, has been one of the strategies used in developing high cup quality coffee. Based on the flavor profile we have assayed, it was found that the ratio of caffeine to sucrose is higher than desirable for good cup quality. It remains to be seen whether there are polymorphisms for known enzymes and proteins that regulate the levels of these compounds among our genotypes. Such data can be obtained by careful examination of our GBS-SNP data, and the utility of this information will be further extended by using them as genetic markers for breeding. By altering this design, the desirable flavor profile will be achieved. It will be of great interest to characterize those genotypes conferring relatively low caffeine, and high sucrose, together with other important quality precursors to determine the cause of such variation and identify some important quality trait markers.

## Figures and Tables

**Figure 1 plants-08-00425-f001:**
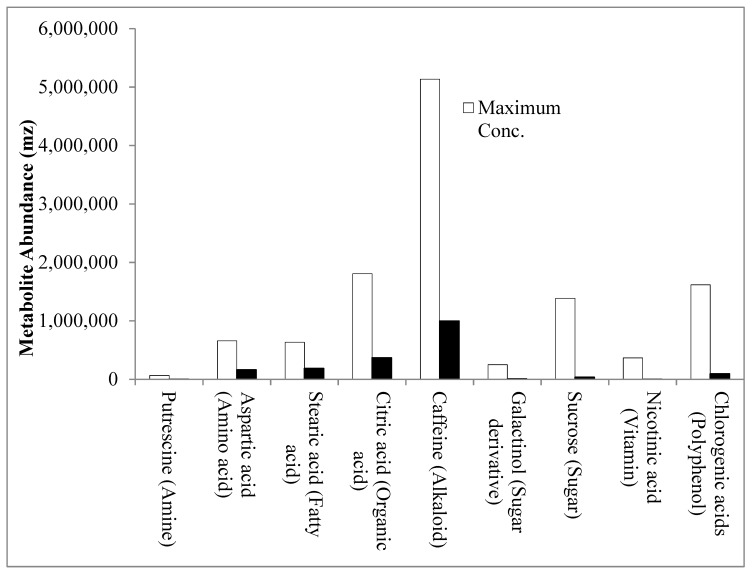
Variation in the relative abundance of coffee metabolites across genotypes. Metabolites that occur with maximal abundance within each of the eight main metabolite classes are indicated by the white bars. Also shown (indicated as a black bar) is the minimal abundance of that metabolite across genotypes.

**Figure 2 plants-08-00425-f002:**
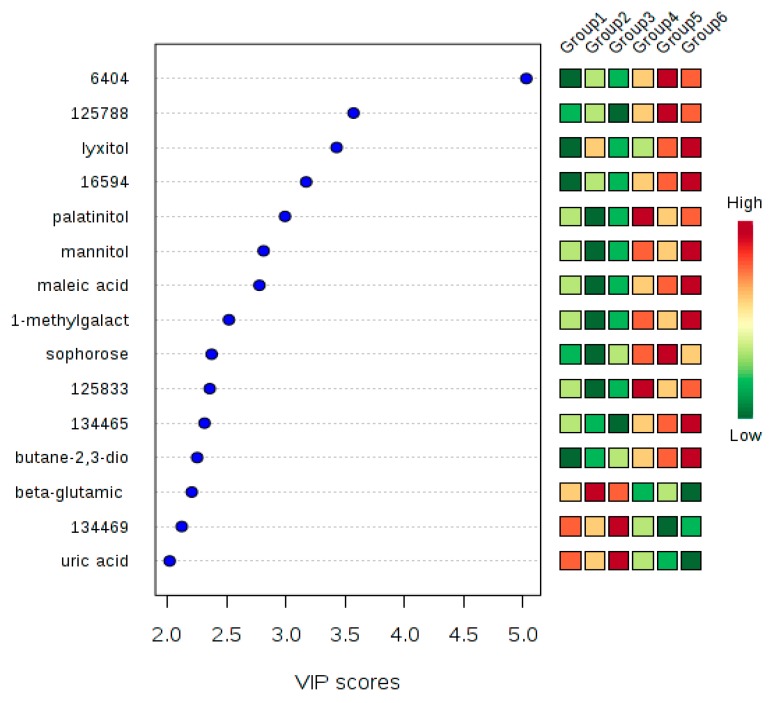
Potential metabolomic markers differentiating ‘Niaouli’ (genotypes groups 1–3) from ‘Kouilou’ (groups 4–5) and ‘Java’ (group 6). The variable importance in projection (VIP) scores on the x-axis provide an estimate of the contribution of a given predictor (metabolites shown on the y-axis) to the Partial Least Square (PLS) regression above 2. The higher the VIP score, the better the metabolite is as a predictor of the discrimination among genotypes.

**Figure 3 plants-08-00425-f003:**
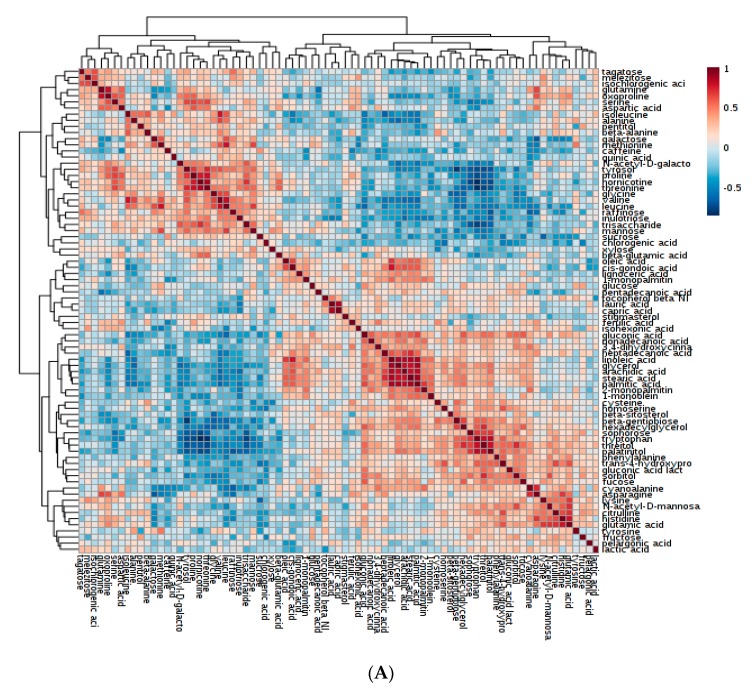
Heatmaps showing metabolite-to-metabolite correlation patterns. Two maps were developed for clarity (**A**). Heatmap showing the correlation among sugars, amino acids, polyphenols and fatty acids. (**B**). Heatmap showing the correlative pattern among organic acids, sugar derivatives, amines and vitamins.

**Figure 4 plants-08-00425-f004:**
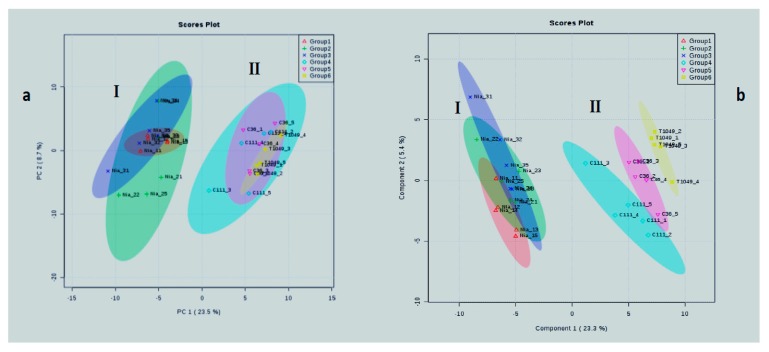
Multivariate classification of Nigerian *Coffea canephora* coffee genotypes. (**a**) Principal Component Analysis (PCA) and (**b**) Partial Least Squared (PLS) Analysis. PLS is a supervised method that minimizes within group variability and maximizes intergroup variability to achieve the greatest separation (discrimination) among groups.

**Figure 5 plants-08-00425-f005:**
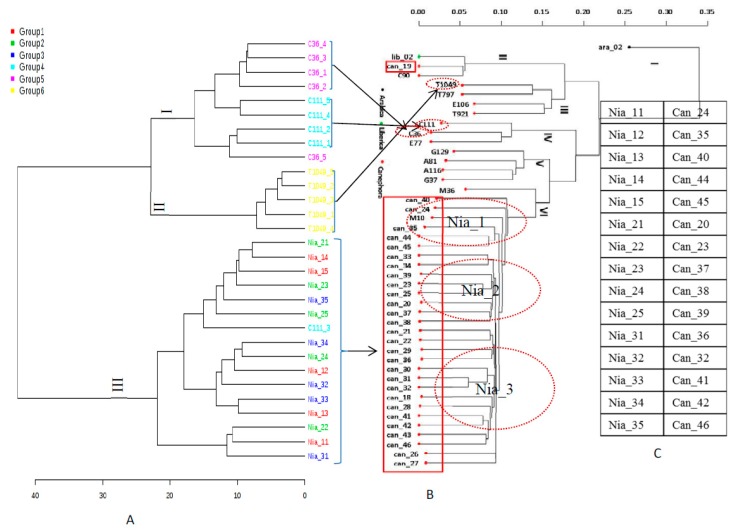
Comparing the metabolite profiles and Single Nucleotide Polymorphism (SNP) diversity of *C canephora*. (**A**) Hierarchical clustering of the metabolomics data derived from 30 coffee samples. (**B**) Hierarchical clustering of Single Nucleotide Polymorphism-Genotype-By-Sequencing data derived from 47 coffee samples, including those used in the metabolomics analysis. Dotted red circles denote genotypes used for metabolomics analysis. The red rectangular box contains the ‘Niaouli’ genotypes. (**C**) This table matches the ‘Niaouli’ genotypes in (**A**) to those in (**B**), where Nia_11 is the same as Can_24.

**Figure 6 plants-08-00425-f006:**
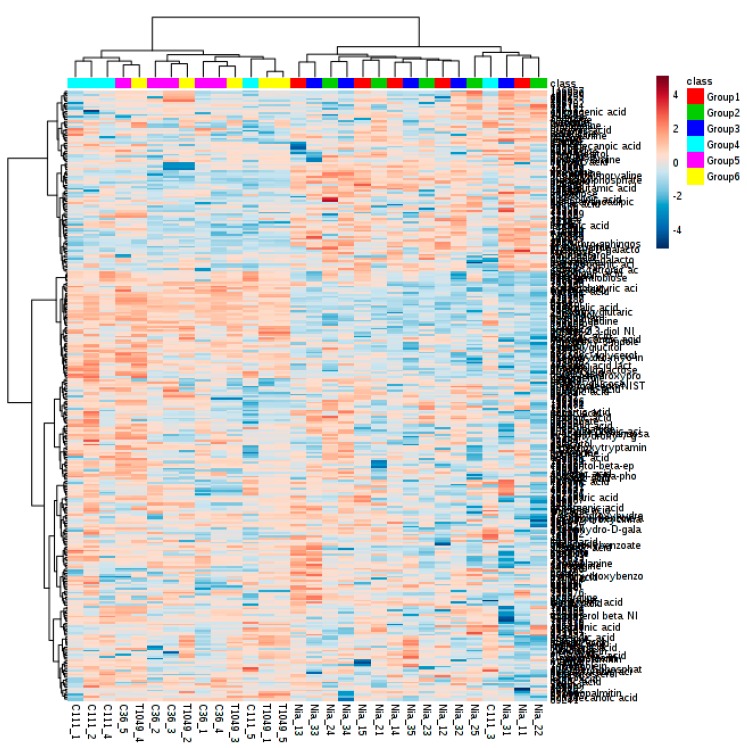
The hierarchical clustering analysis (HCA) showing metabolite diversity and intensity levels in relation to genotypes. Dark brown (4) represents high intensity, while dark blue (−4) represents very low intensity.

**Figure 7 plants-08-00425-f007:**
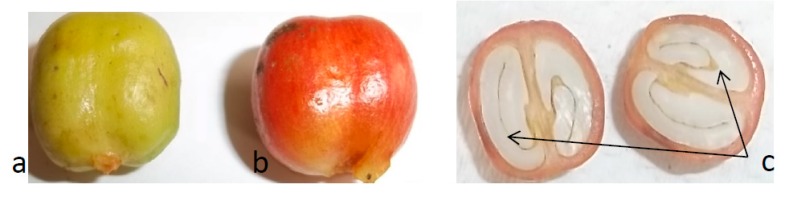
*C. canephora* coffee beans. Two maturity stages of the beans, (**a**) unripe, and (**b**) ripe are shown. Note, only the ripe beans were used for the analysis. (**c**) The endosperm, which was the portion of the coffee bean dissected and used to extract metabolites for chemical analysis.

**Table 1 plants-08-00425-t001:** Metabolomic markers that could be used to differentiate ‘Java/Kouilou’ from ‘Niaouli’ genotypes as detected by one-way ANOVA and post-hoc analyses (*f* > 4.0; *p* < 0.01; −Log_10_ (*p*) > 2.0; FDR < 0.05). Fisher’s LSD identified ‘Java/Kouilou’ to be higher in content in the metabolites listed here compared to ‘Niaouli.’ The *f*-value is derived from the *F*-statistic test for significance, the *p*-value is used to test variability between two groups, −LOG_10_(*p*) determines the significant levels, such as *, **. and ***. The False discovery rate (FDR) assists in testing significant result from *p*-value.

Metabolomic Markers	*f*. Value	*p*. Value	−Log10(p)	FDR
6404	37.478	1.41 × 10^−10^	9.8511	4.30 × 10^−8^
125788	23.897	1.35 × 10^−8^	7.8688	2.06 × 10^−6^
Citramalic acid	21.799	3.29 × 10^−8^	7.4826	2.96 × 10^−6^
Tryptophan	21.428	3.88 × 10^−8^	7.4112	2.96 × 10^−6^
34007	14.774	1.17 × 10^−6^	5.9327	4.29 × 10^−5^
Palatinitol	14.587	1.30 × 10^−6^	5.8846	4.29 × 10^−5^
134465	14.39	1.47 × 10^−6^	5.8333	4.29 × 10^−5^
2193	14.304	1.55 × 10^−6^	5.8109	4.29 × 10^−5^
Erythritol	13.379	2.74 × 10^−6^	5.5625	6.96 × 10^−5^
134464	13.05	3.38 × 10^−6^	5.4711	7.36 × 10^−5^
Threitol	12.92	3.68 × 10^−6^	5.4344	7.48 × 10^−5^
1-methylgalactose NIST	12.663	4.35 × 10^−6^	5.3615	8.29 × 10^−5^
127358	12.044	6.58 × 10^−6^	5.1815	0.0001
Gluconic acid	11.973	6.91 × 10^−6^	5.1606	0.0001
2-hydroxyglutaric acid	11.239	1.15 × 10^−5^	4.9383	0.000146
3182	11.072	1.30 × 10^−5^	4.8861	0.000152
Maleic acid	10.247	2.39 × 10^−5^	4.6219	0.000241
Sophorose	10.215	2.45 × 10^−5^	4.6114	0.000241
16594	9.5222	4.18 × 10^−5^	4.3787	0.000399
4850	9.3508	4.79 × 10^−5^	4.3195	0.000443
125830	8.252	0.000119	3.9236	0.00107
Butane-2,3-diol NIST	8.0335	0.000144	3.8414	0.001253
1,2-anhydro-myo-inositol NIST	8.0039	0.000148	3.8301	0.001253
Pseudo uridine	7.6369	0.000205	3.6888	0.001688
6-deoxyglucitol	7.2396	0.000294	3.5316	0.00236
Mannitol	7.0457	0.000352	3.4532	0.002755
102728	6.4686	0.000612	3.2131	0.004446
133018	5.6757	0.001363	2.8655	0.009037
Hexadecylglycerol NIST	4.8794	0.003208	2.4938	0.019967
Arachidic acid	4.5156	0.004832	2.3159	0.026794
Sorbitol	4.4837	0.005011	2.3001	0.02724
trans-4-hydroxyproline	4.0293	0.008518	2.0697	0.042538
beta-gentiobiose	3.8623	0.010403	1.9829	0.048073
Fisher’s Least Square Difference (LSD) identified ‘Niaouli’ to be higher in content in the metabolites listed here compared to ‘Java/Kouilou’.				
Threonine	16.164	5.27 × 10^−7^	6.2779	3.22 × 10^−5^
Uric acid	15.162	9.30 × 10^−7^	6.0315	4.29 × 10^−5^
Nornicotine	13.222	3.03 × 10^−6^	5.5189	7.10 × 10^−5^
Adipic	12.564	4.64 × 10^−6^	5.3331	8.33 × 10^−5^
17094	12.205	5.90 × 10^−6^	5.229	9.47 × 10^−5^
Pentitol	11.778	7.90 × 10^−6^	5.1025	0.000109
5-hydroxynorvaline NIST	11.414	1.02 × 10^−5^	4.992	0.000135
Tyrosol	11.076	1.30 × 10^−5^	4.8875	0.000152
Proline	10.719	1.68 × 10^−5^	4.7747	0.00019
Methanolphosphate	10.547	1.91 × 10^−5^	4.7195	0.000201
Trisaccharide	5.7878	0.001214	2.9159	0.008225
110009	4.2989	0.006203	2.2074	0.032621
Isocitric acid	4.114	0.007704	2.1133	0.039825
Fisher’s LSD identified ‘Java’ to be higher in content in the metabolites listed here compared to ‘Kouilou’				
Lyxitol	12.382	5.24 × 10^−6^	5.2807	8.88 × 10^−5^
Glycerol	5.4388	0.001748	2.7574	0.01111

**Table 2 plants-08-00425-t002:** Metabolites showing similar correlation pattern with sucrose and caffeine (*p* < 0.05).

Metabolite Identified as Markers	Correlation Coefficient (*r*^2^)	
	Sucrose	Caffeine
Caffeine	0.4079	-
16548	0.4044	0.5182
6404	−0.3705	-
Palatinitol	−0.4961	-
68	0.4991	0.4502
Tryptophan	−0.4599	-

**Table 3 plants-08-00425-t003:** Genotypes grouped based on the relative levels of the key metabolites identified from hierarchical analysis (Figure 6).

Metabolites	Metabolite Levels				
	Very High	High	Medium	Low	Very Low
Caffeine		Nia_24, Nia_25		Nia_15, Nia_14	
CGA		Nia_25		Nia_14	C111_2
Sucrose		Nia_25, Nia_22, Nia_15, Nia_14	Nia_22	Nia_24	C36_5
Quinic acid		Nia_25, Nia_31, Nia_34	C36_1	Nia_14	
Butane-2,3 diol		C36_2		All Nia except Nia_25	Nia_22
Saccharic acid		Nia_15		Nia_22, Nia_11, Nia_31	
Ferulic acid		Nia_33			
Tryptophan		C36_5		Nia_24	
Putrescine	C36_5			Nia_22, Nia_25, Nia_32, Nia_33	C111_5
Proline		Nia_33			C111_3

Very high (4 and above); high (2 to 4); medium (0 to 2); low (0 to −2); and very low (−2 to −4).

**Table 4 plants-08-00425-t004:** Relative levels of metabolites that contribute to coffee quality based on published data.

	Relative Metabolite Levels		
Coffee Type and Perceived Quality	High	Low	Source
Palm civet (Superior)	Citric acid, malic acid, and glycolic acid	Quinic acid, caffeine, and caffeic acid	[21]
*Coffea arabica* (Good)	Sucrose, triglyceride, and threonine, proline	Caffeine, chlorogenic acid, aminobutyric acid (GABA), quinic acid, choline, acetic acid, and fatty acid	[9,19,27]
*C. canephora* (Poor)	Caffeine and chlorogenic acid	Sucrose	[45,46]

**Table 5 plants-08-00425-t005:** Sample genotypes and symbols.

Variety	‘Niaouli’			‘Kouilou’		‘Java Robusta’
Group/Genotype	Group 1 (Nia_1)	Group 2 (Nia_2)	Group 3 (Nia_3)	Group 4 (C111)	Group 5 (C36)	Group 6 (T1049)
**Sample Symbols**	Nia_11	Nia_21	Nia_31	C111_1	C36_1	T1049_1
	Nia_12	Nia_22	Nia_32	C111_2	C36_2	T1049_2
	Nia_13	Nia_23	Nia_33	C111_3	C36_3	T1049_3
	Nia_14	Nia_24	Nia_34	C111_4	C36_4	T1049_4
	Nia_15	Nia_25	Nia_35	C111_5	C36_5	T1049_5

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
