# Peer review of "Gas Chromatography-Mass Spectrometry and Single Nucleotide Polymorphism-Genotype-By-Sequencing Analyses Reveal the Bean Chemical Profiles and Relatedness of Coffea canephora Genotypes in Nigeria"

_plants, 2019, doi:10.3390/plants8100425_

Round 1

Reviewer 1 Report

In this work, Anagbogu et al. describe the application of GC/MS-based metabolite profiling and SNP genotyping to differentiate various Nigerian genotypes of coffee (C. canephora) with the aim of identifying genotypes with optimum organoleptic characteristics to be incorporated in breeding program for quality improvement. The manuscript is clear and well written, and results are of interest for readers of the Plants journal. Below some minor suggestions:

Did authors employ pure standards to confirm the identity of discriminant metabolites analyzed by GC-MS? Correct numbering of Figures and Tables. Correct Figure 2: caffeine is not a phenolic acid, and myo-inositol is not a vitamin. Correct sentence in line 108: “A total of 340 untargeted features were detected”

Reviewer 2 Report

The authors present a paper which deals with GCMS metabolomic and comparative genotype sequencing analysis of Coffea canephora genotypes in Nigeria. I found this an interesting paper and there are a lot of results presented. The work is new and should be published in my opinion.

The table and figure numbers do not tally within the manuscipt in its current format.

Could the authors find a way of summarising previous work on desirerable coffee metabolites from e.g. C. arabica in tabular or figure form for comparison with their work. It is relatively hard to determine what are the major differences and what would be the aim for improvements in genetic modification/breeding.

Could the authors comment on the likely success in altering the major metabolites and how the metabolomic analysis would contribute to the ongoing alteration of flavour profile.

Could the authors comment on the repeatability of the metabolomic analysis and to a lesser extent the genetic sequencing. How representative is it of the compounds in that specific genotype vs. the natural variation of flavour profile by growing conditions and methodological differences. for example what variation is there within the experimental measurements with GC-MS. How consistent are the measurements likely to be between coffee grown in differ suppose I'm trying to understand better the variation if this is then used as a measure of success in altering desireable metabolites/adjusting ratios or reducing undesireable metabolites.  
